# Translating Biomarker Discovery: From Bench to Bedside in Dry Eye Disease

**DOI:** 10.3390/ijms26178556

**Published:** 2025-09-03

**Authors:** Jeremy Jones, Kyla Frenia, Julia Gelman, Maria Beatty, Melody Zhou, Levin Ma, Sean Pieramici, Noah Eger, Deepinder Dhaliwal, Leanne T. Labriola, Kunhong Xiao

**Affiliations:** 1Pittsburgh Research Institute, Sewickley, PA 15143, USA; jeremy@pghtrials.com (J.J.); gelmanjulia2@gmail.com (J.G.); maria@pghtrials.com (M.B.); 2Department of Bioengineering, Swanson School of Engineering, University of Pittsburgh, Pittsburgh, PA 15260, USA; frenia@innsightech.com; 3Department of Biomedical Engineering, College of Engineering, Carnegie Mellon University, Pittsburgh, PA 15213, USA; melody.z9690@gmail.com (M.Z.); llmar1000000@gmail.com (L.M.); 4Southwestern PA Eye Center, Corneal Services, Washington, PA 15301, USA; spieramici@gmail.com; 5Eger Eye Group, Optometry, Coraopolis, PA 15108, USA; noahod@verizon.net; 6University of Pittsburgh Medical Center, Corneal Services, Pittsburgh, PA 15260, USA; dhaliwaldk@upmc.edu; 7Center for Proteomics & Artificial Intelligence, Allegheny Health Network Cancer Institute, Pittsburgh, PA 15212, USA; 8Center for Clinical Mass Spectrometry, Allegheny Health Network Cancer Institute, Pittsburgh, PA 15212, USA

**Keywords:** dry, diagnostics, eye, inflammation, lactoferrin, matrix metalloproteinase-9, osmolarity, osmometry

## Abstract

Dry Eye Disease (DED) is a complex, multifaceted ocular disease characterized by tear film instability and inflammation. It can sometimes be elusive to identify the type of DED in patients, given the overlapping symptoms with other conditions like allergies and the multitude of stimuli that might trigger DED onset. There is also difficulty due to limitations on the diagnostic testing available to clinicians, as poor reliability and a lack of standardization plague accurate diagnoses. Identified biomarkers can help identify DED pathophysiology and category, and these include molecular biomarkers like matrix metalloproteinase-9 (MMP-9), cytokines, lactotransferrin, and lacritin, as well as functional biomarkers such as tear osmolarity. Diagnostic tools, such as the InflammaDry and I-Pen Tear Osmolarity System, also now allow for point-of-care measurement of select biomarkers, including MMP-9 and osmolarity. Nonetheless, there remains a critical need for additional, reliable, and accurate diagnostic devices to better aid in the diagnosis and management of DED. This review uniquely combines a review on the current understanding of various biomarkers with an overview of the emerging technologies available to healthcare providers, aiding in better-informed diagnosis and treatment of DED.

## 1. Introduction

Dry Eye Disease (DED) is a prevalent ocular condition characterized by tear film instability and ocular surface inflammation, ultimately leading to cellular damage, discomfort, and visual disturbances [1]. The tear film, comprising an aqueous, lipid, and mucin layer, is essential to ocular surface health and visual acuity. Tear film instability can arise via several mechanisms, including reduced tear production, excessive tear evaporation, and changes in mucin secretion [2]. Often, individuals experiencing tear film instability experience a mixture of underlying etiologies. Contributors to tear film instability include systemic conditions (e.g., autoimmune disorders), environmental factors (e.g., digital eye strain, air pollution), lifestyle choices (e.g., poor nutrition, contact lens use), and geographic influences (e.g., low humidity, high altitude) [1]. Due to the multifaceted nature of DED, diagnosis remains challenging, especially because DED symptoms often overlap with other ocular conditions and can vary in clinical presentations. Molecular biomarkers have emerged as a reliable and objective metric for the diagnosis and screening of DED. These markers also provide new insights into inflammatory pathways, lacrimal gland dysfunction, and tear film dynamics. This review synthesizes current knowledge on DED molecular biomarkers, focusing on their role in advancing mechanistic understanding. An improved understanding of DED-relevant biomarkers and how to best use the tools currently available to quantify these biomarkers will aid in advancing more accurate diagnosis of DED in patients, leading to treatment specific to DED subtype and improved outcomes in the future.

## 2. Pathophysiology of DED

DED is complex, often driven by a combination of underlying mechanisms. Often, tear instability presents as the central initiating event, beginning with a breakdown in tear film. This is caused either by a decrease in tear secretion, common in aqueous-deficient DED, or by an increase in tear evaporation, more common in evaporative DED. Once the tear film is compromised, the remaining tear fluid is more concentrated with salts and other solutes, leading to a state of hyperosmolarity. The resulting damage to the ocular surface initiates a self-perpetuating cycle of inflammation, which makes symptoms worse over time and prevents the disease from resolving without intervention [2,3,4,5,6,7]. The initial stress and damage to the ocular surface epithelial cells leads to the production and release of pro-inflammatory cytokines and matrix metalloproteinases, most notably including matrix metalloproteinase-9 (MMP-9), which compromises the integrity of the corneal epithelium (Figure 1) [5]. This inflammatory cascade is part of a normal bodily inflammatory response, but the chronic presence of these cytokines results in further ocular surface damage and worsens the tear film’s state of dysregulation. The chronic inflammation not only impacts the ocular surface but concurrently impairs the lacrimal glands as well, leading to reduced tear production. It also damages the meibomian glands, which alters lipid composition and further increases tear evaporation. This further destabilizes the tear film and reinforces the inflammatory environment of the ocular surface [6]. Neurosensory abnormalities resulting from this inflammation, such as impairment of corneal nerve function and increased nociceptive signaling, can further lead to significant patient discomfort [7]. This complex interplay among tear film instability, immune activation, and neurosensory feedback forms a vicious cycle that sustains the chronic nature of DED. Understanding these interrelated mechanisms is essential for identifying targeted biomarkers and developing more effective, personalized therapeutic strategies for DED.

## 3. Molecular Biomarkers of DED

Biomarkers play an emerging role in improving the diagnosis, subtyping, and management of DED, a condition that typically presents with indistinct symptoms and overlapping clinical features. Traditional diagnostic methods, such as Schirmer’s test or tear break-up time, are limited by variability and lack of sensitivity, emphasizing the need for objective molecular markers [8,9,10]. In recent years, a diverse array of candidate biomarkers—including small molecules, proteins, and lipids—have been identified in the tear film (Table 1) [11,12,13,14,15,16,17,18]. Each marker offers unique insights into the underlying pathophysiology of DED, reflecting inflammation, glandular dysfunction, tear composition, or neurosensory abnormalities. Identification of these markers has elucidated the underlying mechanisms of DED, including key processes such as the recruitment of cytotoxic T cells and CD4+ T-helper cells, activation of antigen-presenting cells, and release of inflammatory mediators such as MMP-9, Tumor Necrosis Factor alpha (TNF-α), Interleukin-1 (IL-1), and Interleukin-6 (IL-6) [2,5,19,20,21,22]. These markers contribute to ocular surface damage and disease chronicity, making them prime candidates for both diagnostic and therapeutic applications. The following sections provide an overview of key biomarker categories and their clinical relevance, with particular emphasis on MMP-9 and other inflammation-related markers closely associated with DED.

### 3.1. Inflammatory Markers

Inflammation is a key process that perpetuates DED and contributes to many of the symptoms of discomfort experienced by patients daily. The hyperosmolarity that initiates DED, along with mechanical trauma to the cornea due to reduced lubrication of the ocular surface, both trigger and sustain inflammation. This inflammatory response can further exacerbate DED by damaging the ocular surface and lacrimal glands, leading to a chronic state of injury that continuously stimulates immune and inflammatory pathways. Because inflammation is a primary driver of DED progression, inflammatory proteins are often strong biomarkers of disease. These markers frequently correlate with clinical signs, patient-reported symptoms, and disease severity.

#### 3.1.1. Matrix Metalloproteinases and MMP-9

MMP-9 is the most well-known of the molecular biomarkers for DED, having been successfully translated to clinical use through the InflammaDry™ device (QuidelOrtho, San Diego, CA, USA) [23,24,25,26]. As a zinc-dependent endopeptidase, MMP-9 degrades extracellular matrix components, including corneal epithelial tight junction proteins, leading to barrier disruption and accelerated epithelial cell shedding [24]. Elevated MMP-9 levels in tears are strongly correlated with DED severity, shorter tear break-up times, and compromised corneal integrity [27]. Prior clinical studies have supported the implication of MMP-9 in DED patients. One study recruited forty-six patients with newly diagnosed DED and 18 control subjects. They collected 1 microliter of unstimulated tear fluid from each patient and analyzed them for MMP-9 levels. The study found that MMP-9 was significantly higher in DED patients than in the control patients, and the MMP-9 levels even showed to be significantly correlated to DED patients in this study [28]. Mechanistically, MMP-9 is upregulated by pro-inflammatory cytokines like Interleukin-1β (IL-1β) and TNF-α, which are released by activated immune cells, including cytotoxic T cells and CD4+ T-helper cells [19]. This creates an inflammatory feedback loop that exacerbates tissue damage. Due to MMP-9’s responsiveness to inflammatory stimuli and its direct impact on ocular surface health, MMP-9 serves as a valuable biomarker for investigating DED pathophysiology and guiding targeted therapeutic strategies.

MMP-9 is also considered an objective biomarker for DED and is correlated with ocular surface integrity [24,29,30,31]. An increased concentration of MMP-9 is typically associated with a shorter tear break-up time, which serves as a measure of tear film stability. This information is especially useful in clinical settings, as a reduced tear break-up time is reflective of an unstable tear film, making it difficult for patients to maintain visual clarity and comfort. Thus, MMP-9 serves as a meaningful marker of inflammation, ocular surface integrity, and functional tear film deficiency in DED. Other matrix metalloproteinases (MPPs) have also been identified as candidate biomarkers of DED, including MMP-2 and MMP-3, although these are less established.

#### 3.1.2. Cytokines and Chemokines

Cellular communication within the immune system is crucial for coordinating a proper response to pathogens and injury. This communication is mediated by a diverse family of small proteins called cytokines, which act as signaling molecules to regulate the behavior of immune cells. Chemokines are a subset of cytokines that play a specific role in guiding the migration of these cells to sites of inflammation or infection. Cytokines and chemokines both play a significant role in recruiting immune cells to the ocular surface in DED, contributing to the perpetuation of inflammation and corneal damage in DED.

IL-1β is a key pro-inflammatory cytokine involved in ocular surface inflammation, particularly in DED [32,33]. It is primarily produced by activated innate immune cells as an inactive prohormone and can be enzymatically cleaved intracellularly or extracellularly to become active. Extracellular cleavage can be performed by a variety of enzymes, most notably including MMP-9, suggesting IL-1β may play a significant role in the self-perpetuating inflammatory cycle that characterizes DED. IL-1β is widely considered a master regulator of both local and systemic inflammation, playing a pivotal role in the development and progression of acute and chronic inflammation. Although it is present in healthy tear fluid, it exists predominantly in its inactive form. Some studies have shown IL-1β to be present at higher levels in DED tear fluid compared to healthy tear fluid. IL-1β could pose potential as a biomarker for DED, especially considering its role in regulating inflammatory events and association with MMP-9.

Interleukin-6 (IL-6) is a pleiotropic cytokine also involved in DED, possessing both pro- and anti-inflammatory characteristics. IL-6 is consistently found at significantly elevated concentrations in the tears of patients diagnosed with DED when compared to healthy control individuals [34]. This consistent elevation underscores its potential as a key indicator of disease presence and activity. On the ocular surface, IL-6 is largely produced by corneal epithelial cells in response to DED-like triggers such as hyperosmolarity [21]. Interestingly, IL-6 also plays a significant role in initiating the differentiation of T-helper 17 (Th17) cells, which are known to be critical players in chronic inflammatory diseases and disease relapse [35]. Thus, IL-6 is a strong candidate as a target for therapeutic intervention and as a biomarker, given its potential role in the pathogenesis of DED and elevation in DED tear samples.

TNF-α, a pleiotropic cytokine involved in numerous inflammatory responses, has been found to be elevated in the tear film and ocular surface tissues of patients with DED. Its upregulation contributes to the chronic inflammation characteristic of the disease, affecting both the ocular surface and lacrimal glands [36]. However, while TNF-α plays a critical role in DED pathogenesis and is associated with disease severity, it is not a disease-specific biomarker due to its overlap with broader systemic inflammatory processes, limiting its diagnostic specificity.

Similarly, IgE also plays a significant role in ocular surface inflammation [37,38]. The inflammatory environment brought about by IgE-mediated reactions can compromise the epithelial barrier function. When the epithelial barrier is damaged, increased tear evaporation occurs and irritation persists. Evidently, DED symptoms worsen because of the activity of IgE. While increased IgE levels in tear fluid are not directly associated with DED, they are closely linked to a comorbid condition, allergic conjunctivitis. The symptoms of these two conditions are nearly identical, making it difficult to determine whether ocular surface irritation and dryness are present due to disease or allergic reactions. Simultaneously, researchers are investigating whether an underlying allergy component exists in DED. Moreover, IgE, though elevated in tear fluid during allergic conjunctivitis—a condition often overlapping with DED—primarily reflects allergic comorbidity rather than direct DED pathophysiology, necessitating contextual interpretation to avoid misdiagnosis.

**Table 1 ijms-26-08556-t001:** Key Biomarkers for DED.

Biomarker	Role	Clinical Relevance	Reference
1. Protein Biomarkers			
MMP-9	Degrades extracellular matrix; upregulated during inflammation	Elevated in DED; used in point-of-care test (InflammaDry); marker of ocular surface inflammation	[23,24,25,26,27,29,30,31]
Lactoferrin	Iron-binding glycoprotein with antimicrobial and anti-inflammatory properties	Decreased in aqueous-deficient DED; indicates lacrimal gland dysfunction	[39,40,41,42]
Lacritin	Tear glycoprotein that promotes epithelial cell survival, autophagy, and secretion	Deficient in aqueous-deficient DED; shown to restore tear secretion and corneal integrity in preclinical models	[43,44,45,46,47]
Lysozyme	Antimicrobial enzyme secreted by lacrimal glands	Reduced levels suggest impaired tear secretion	[48,49,50,51,52]
Lipocalin-1	Stabilizes the tear film lipid layer	Altered levels associated with tear film instability	[43,53,54,55,56]
MUC5AC	Secreted gel-forming mucin from conjunctival goblet cells	Decreased in DED, especially in mucin-deficient or Sjögren’s syndrome cases	[57,58,59,60,61,62,63,64,65,66,67]
HLA-DR	Major histocompatibility complex class II molecule	Upregulated in conjunctival epithelial cells; marker of immune activation	[68,69,70,71,72]
2. Cytokines and Chemokines			
IL-1β, IL-6, TNF-α	Pro-inflammatory cytokines	Elevated levels in tears of DED patients; drive ocular surface inflammation	[22,73,74,75,76,77,78,79,80,81]
IL-8 (CXCL8)	Neutrophil chemoattractant	Reflects active inflammation and epithelial damage	[5,17,75,82,83,84]
IFN-γ	Activates immune response, especially Th1-mediated	Linked to goblet cell loss and mucin downregulation	[17,75,82,84,85]
CCL5 (RANTES)	Recruits T cells	Found in increased levels in tears and conjunctiva of DED patients	[5,86,87]
3. Lipid Biomarkers			
Meibum Lipids (e.g., wax esters, cholesterol esters)	Maintain tear film stability and reduce evaporation	Altered composition in Meibomian Gland Dysfunction (MGD) contributes to evaporative DED	[88,89,90,91,92,93,94]
Phospholipids, sphingolipids	Inflammatory signaling molecules	Lipidomics has revealed dysregulated lipid profiles in DED associated with inflammation	[95,96,97,98,99,100,101,102]
4. Metabolites and Small Molecules			
Lactate, Urea	Indicators of metabolic stress	Elevated levels found in tear fluid of DED patients	[103,104,105]
Glutamate, Glutamine	Linked to oxidative stress and inflammation	Altered profiles can distinguish DED subtypes	[106,107,108,109]
Reactive oxygen species (ROS)	Oxidative stress marker	Associated with cellular damage in DED pathogenesis	[110,111,112,113]
5. Nucleic Acid Biomarkers (Genomic/Epigenomic)			
MicroRNAs (e.g., miR-146a, miR-155)	Post-transcriptional gene regulation of inflammation	Dysregulated in tears and conjunctiva; potential non-invasive biomarkers for DED diagnosis and subtype stratification	[114,115,116,117,118,119]
HLA gene polymorphisms	Immune response genes	Certain variants associated with Sjögren’s syndrome and autoimmune-related DED	[120,121,122,123]
6. Functional and Imaging Biomarkers			
Tear Osmolarity	Measures tear solute concentration	Elevated (>308 mOsm/L) in DED; reproducible marker for severity	[124,125,126,127,128,129]
Corneal Sensitivity	Assesses corneal nerve function and ocular surface integrity	Reduced in DED; associated with neurosensory abnormalities and disease severity	[130,131,132,133]
Tear Break-Up Time (TBUT/NITBUT)	Measures tear film stability	Decreased in DED, especially in evaporative forms	[134,135,136,137,138]
Meibography	Visualizes meibomian gland structure	Gland dropout seen in MGD-related DED	[139,140,141,142]
In vivo confocal microscopy (IVCM)	Assesses corneal nerves and immune cells	Reveals corneal nerve loss or dendritic cell activation in DED	[143,144,145,146,147,148]

### 3.2. Lacrimal Gland Protein Markers

Differential expression of lacrimal gland-derived proteins, which reflect alterations in tear fluid synthesis and secretion, is also commonly reported as a marker of DED. These proteins may indicate dysfunction or damage to the lacrimal glands, a hallmark of certain DED subtypes, particularly aqueous-deficient DED. Changes in expression levels can signal impaired tear production, glandular inflammation, or structural degeneration, all of which contribute to tear film instability and ocular surface stress. As such, lacrimal gland proteins serve not only as potential biomarkers of disease but also as indicators of underlying pathophysiological mechanisms driving DED progression.

#### 3.2.1. Lactoferrin

Lactoferrin (LTF or LF), also known as lactotransferrin, is an 80 kDa multifunctional glycoprotein. It is one of the most abundant proteins in tear fluid and is a key biomarker for diagnosing DED [40,41]. In the tear film, lactoferrin plays a crucial role in maintaining ocular surface health through its antimicrobial, anti-inflammatory, and antioxidant properties. A marked reduction in tear lactoferrin levels is commonly seen in patients with DED, particularly in the aqueous-deficient subtype caused by lacrimal gland dysfunction. This decline correlates with reduced tear production and increased ocular surface damage. Measuring lactoferrin levels provides a non-invasive and reliable method to differentiate between types of DED, such as Sjögren’s syndrome-associated DED or other non-Sjögren’s variants [39]. Emerging diagnostic technologies, such as photo-detection devices and microfluidic assays, have enhanced precision and accessibility in lactoferrin measurement, facilitating more personalized treatment strategies. These advancements underscore lactoferrin’s value as a diagnostic and prognostic biomarker for DED, facilitating targeted therapies to mitigate inflammation and restore tear film homeostasis.

#### 3.2.2. Lysosome

Lysozyme (LYZ) is a glycoside hydrolase that plays a critical antimicrobial role in the tear film, helping to protect the ocular surface from bacterial invasion [42]. Like lactoferrin, it is produced by the lacrimal gland and secreted into the aqueous layer of the tear film, contributing to the innate immune defense of the ocular surface. In the context of DED, decreased concentrations of lysozyme are commonly associated with aqueous-deficient subtypes and may indicate underlying lacrimal gland dysfunction [44,45,46]. Because of its high abundance in healthy tear fluid and its sensitivity to changes in glandular output, lysozyme has been proposed as a useful biomarker for assessing tear film integrity and lacrimal gland health. When evaluated alongside other tear proteins such as lactoferrin and lipocalin-1, lysozyme can contribute to a more comprehensive characterization of the ocular surface environment and aid in the stratification of DED subtypes. Moreover, changes in lysozyme levels over time may provide insight into disease progression or response to therapeutic intervention.

### 3.3. Lipids

The lipid layer of the tear film plays a crucial role in maintaining ocular homeostasis, primarily by reducing the rate of tear evaporation and preserving tear film stability. Secreted predominantly by the meibomian glands, this outermost layer forms a barrier that minimizes fluid loss from the aqueous layer beneath it. Disruption or deficiency of the lipid layer, as seen in meibomian gland dysfunction (MGD), leads to increased tear evaporation, tear film instability, and hyperosmolar stress—key drivers of evaporative DED. Consequently, alterations in the composition or integrity of the lipid layer are both contributors to disease pathogenesis and potential targets for biomarker discovery and therapeutic intervention.

#### 3.3.1. Omega-6 and Omega-3 Fatty Acids

The Omega-6 to Omega-3 fatty acid ratio reflects the balance of pro-inflammatory and anti-inflammatory lipid precursors in the tear film. Omega-6 polyunsaturated fatty acids (PUFAs), such as arachidonic acid (AA), give rise to inflammatory mediators, while Omega-3 PUFAs, like DHA and EPA, are precursors to pro-resolving, anti-inflammatory molecules [149,150]. This ratio is influenced by systemic diet and local lipid metabolism on the ocular surface, impacting the overall inflammatory milieu of the tear film. In the context of DED, an elevated Omega-6 to Omega-3 ratio is frequently observed, indicative of a pro-inflammatory state that perpetuates the cycle of ocular surface damage and tear film instability. Due to its direct link to inflammation, this ratio has been proposed as a valuable biomarker for assessing the inflammatory component of DED and for monitoring the efficacy of anti-inflammatory treatments. When evaluated in conjunction with other tear film components, the Omega-6 to Omega-3 ratio can contribute to a more nuanced understanding of DED pathophysiology and help guide personalized therapeutic strategies. Moreover, shifts in this ratio over time may provide insight into disease progression or response to interventions targeting inflammation. A recent extension study to the Dry Eye Assessment and Management (DREAM) trial sought to find if discontinuation of Omega-3 supplementation in patients previously given Omega-3 as part of the main DREAM study would yield different outcomes in symptoms and discomfort. 22 patients were randomized to Omega-3 supplements and 21 were given a placebo. The results of the study showed that there was no significant difference in symptom outcomes in the group continuing to take the supplement and the group that discontinued it [151]. This study calls into question the significance that Omega-3 fatty acid supplementation has on improving DED. The study is limited by the small cohort size, however, and further research needs to be done to fully evaluate the impact Omega-3 fatty acids may have in DED.

#### 3.3.2. O-acyl-ω-hydroxy Fatty Acids (OAHFAs)

O-acyl-ω-hydroxy fatty acids (OAHFAs) are a unique class of lipids critical for the structural integrity and function of the tear film lipid layer (TFLL) [99,152]. Produced primarily by the meibomian glands, these specialized lipids contribute significantly to the outermost layer of the tear film, forming a stable barrier that retards evaporation of the underlying aqueous layer. In the context of DED, particularly evaporative subtypes stemming from meibomian gland dysfunction (MGD), decreased concentrations of OAHFAs are commonly associated with increased tear film evaporation and instability. Because of their direct role in maintaining the TFLL’s barrier function and their sensitivity to changes in meibomian gland health, OAHFAs have been proposed as useful biomarkers for assessing evaporative dry eye and meibomian gland function. When evaluated alongside other tear film lipids and structural components, OAHFAs can contribute to a more comprehensive characterization of the tear film’s evaporative resistance and aid in the stratification of DED subtypes. Changes in OAHFA levels over time may also provide important insights into disease progression or response to therapeutic interventions targeting MGD.

#### 3.3.3. Diesters (DiEs)

Diesters (DiEs), specifically Type I and Type II Diesters, are prominent components of the tear film lipid layer (TFLL), playing a crucial role in its physical properties and stability [153,154]. These complex lipids are synthesized by the meibomian glands and are essential for forming a uniform and stable lipid monolayer on the aqueous tear surface, which is vital for preventing rapid tear evaporation and ensuring smooth blinking. In the context of DED, particularly in cases linked to meibomian gland dysfunction (MGD), qualitative and quantitative alterations in diester profiles are frequently observed, contributing to tear film instability and increased evaporation. Because of their significant contribution to the TFLL’s structure and function and their sensitivity to meibomian gland health, diesters have been proposed as valuable biomarkers for assessing tear film quality and meibomian gland function in DED. When evaluated in conjunction with other tear film lipids like OAHFAs and meibomian gland expression, diester analysis can contribute to a more comprehensive characterization of the tear film’s evaporative properties and aid in the diagnosis and subtyping of DED.

### 3.4. MicroRNAs (miRNAs)

miRNAs are small non-coding RNAs that regulate gene expression and have recently emerged as pivotal biomarkers in DED pathogenesis [117,155]. Research led by Pflugfelder at Baylor College of Medicine, in conjunction with insights from the 2024 TFOS sessions, has elucidated the role of miRNAs in DED’s inflammatory processes. Specific miRNAs, such as miR-204, are upregulated in the conjunctival epithelium and tear film of DED patients, modulating inflammatory pathways by targeting receptors like Toll-like receptors (TLRs) and the TNFR superfamily. These miRNAs regulate the expression of pro-inflammatory cytokines, including IL-1β, TNF-α, and Interferon gamma (IFN-γ), which exacerbate ocular surface inflammation and lacrimal gland dysfunction. Advanced molecular techniques, such as quantitative PCR and RNA sequencing, have been instrumental in identifying these miRNA expression profiles, providing a deeper understanding of DED’s molecular mechanisms.

## 4. From Biomarkers to Clinical Diagnosis

Multiple devices are commercially available for rapid, in-clinic use that measure a variety of DED biomarkers and measurements (Table 2). These include a device for MMP-9 measurement (InflammaDry, QuidelOrtho, San Diego, CA, USA), and tear osmolarity measurements (I-Pen, I-MED Pharma, Montreal, QC, Canada, and ScoutPro, Bausch & Lomb, Laval, QC, Canada), esthesiometry (Brill corneal esthesiometer, Brill Engines, Barcelona, Spain), and others. Currently, these devices are typically used in the same patients to gather additional data in patients suspected to have DED, yet they do not have the standalone sensitivity to classify the disease etiology (Figure 2).

### 4.1. Tools for MMP-9 Measurement

The only device currently available to clinicians for MMP-9 measurement is the InflammaDry device. InflammaDry is a point-of-care diagnostic approach designed to detect the inflammatory biomarker MMP-9 in tears for DED. MMP-9 is the most recognizable molecular biomarker for DED. Recent TFOS discussions emphasized MMP-9’s key role in corneal epithelial abnormalities, noting its utility in differentiating DED subtypes due to its association with evaporative DED and MGD [156].

In the clinic, InflammaDry provides a non-invasive method of confirming the presence of ocular surface inflammation and uses lateral flow to detect the concentration of MMP-9 expression. A positive result of two lines means that the MMP-9 concentration is greater than 40 ng/mL. A negative result of only one line indicates an MMP-9 concentration is less than 40 ng/mL [25,26,157,158].

A randomized control trial of 206 patients was performed to assess the specificity and sensitivity of InflammaDry [159]. 143 of the patients presented with the signs and symptoms of DED in clinic, and the other 63 patients were healthy controls. The results showed that InflammaDry had an 85% sensitivity rate (121 of 143 DED patients) and a specificity of 94% (59 of 63 healthy patients). A follow up clinical study sponsored by the company confirmed this finding, citing an 87% sensitivity rate and 97% specificity rate.

While the InflammaDry device has high specificity and sensitivity rates, it can be limited in its role in diagnosing DED. Its ability to measure MMP-9 levels is not a standalone tool; other measurements and observations are often required to confirm a DED diagnosis. MMP-9 levels are known to be elevated as a result of ocular conditions besides DED, highlighting the need for additional data collection when diagnosing. It also does not provide a quantifiable level of MMP-9 concentration in tear film, only giving a positive or negative result based on if the concentration level relative to 40 ng/mL. This is a clinical limitation, as some patients with milder DED may yield MMP-9 levels below the detectable threshold on the InflammaDry. Additionally, monitoring treatment response may be limited here too, as the MMP-9 level cannot be directly measured and tracked. More nuance in MMP-9 levels may be required, especially in low-level or mild dry eye.

Overall, the InflammaDry device can be a reliable tool for aiding in-clinic diagnosis of DED. While limited in standalone diagnostic ability, the device fills a need for accurate measurement of MMP-9 for patients at risk of DED.

### 4.2. Tools for Tear Osmolarity Measurement

Tear film osmolarity, integrally linked to lacrimal gland function, plays a crucial role in maintaining ocular surface health [128,129]. Osmolarity, which refers to the concentration of dissolved particles in the tear film, is meticulously regulated under normal conditions. Tear osmolarity reflects the balance of electrolytes, water, and proteins in the tear film, regulated by lacrimal gland secretion. In DED, dysfunction of the lacrimal gland can lead to hyperosmolarity, often because of reduced aqueous production or excessive evaporation [126]. This hyperosmolarity damages the ocular surface epithelium and triggers inflammatory cascades [160]. These changes further exacerbate the symptoms of DED, creating a self-perpetuating cycle of ocular surface damage and inflammation. Given that elevated osmolarity is a hallmark of DED, measurement of tear osmolarity can provide insights into lacrimal gland dysfunction and tear film dynamics and has emerged as a valuable diagnostic tool for DED. Additionally, understanding and targeting tear film osmolarity could potentially lead to more effective treatments for DED, aiming to restore the delicate osmotic balance necessary for optimal ocular surface health.

One common device used in clinics for tear osmolarity measurement is the I-Pen tear osmolarity system. The I-Pen system offers a rapid, handheld method of measuring tear film osmolarity to aid in diagnosing DED in patients. The device uses a single-use sensor probe that is gently placed on the lower eyelid’s inner surface for approximately two seconds to generate the result. The rapid test displays results immediately in mOsm/L. It requires no anesthesia or sample collection and can be easily performed by trained clinical staff. The I-Pen is very portable and delivers results rapidly, making it a convenient tool for in-clinic use. Comparatively, other similar devices like the Wescor or TearLab are somewhat less portable. Primarily, the differences between these devices are in the collection and analysis methods and in the accuracy of reported results based on prior studies.

A clinical study published in 2021 evaluated the efficacy of the I-Pen in comparison to other standard diagnostic methods. 65 patients were enrolled in the study, with 32 patients presenting with DED pathology and 33 patients presenting with no DED symptoms. Using the osmolarity cutoff of 318 mOsm/L, the study revealed a 90.9% sensitivity and a 90.6% specificity for identifying DED [161]. However, a 2017 in vitro study compared the performance of the I-Med Pharma I-Pen, Wescor 5520 Vapro Pressure Osmometer, and TearLab Osmolarity System, and found that the I-Pen was neither accurate nor precise, especially compared to the other two devices. The study used solutions of known osmolarity to evaluate each device. The Wescor and TearLab devices had correlation coefficient values of r^2^ = 0.98 and r^2^ = 0.96, respectively, while the I-Pen had a value of r^2^ = 0.03. Additionally, its coefficients of variation (CVs) were notably high, ranging from 6.1% to 6.4%. This performance contrasts sharply with the Wescor device (CV = 1.0–1.6%) and the TearLab system (CV = 1.2–2.3%), further supporting a lack of accuracy and precision in the I-Pen device.

Another common device for tear osmolarity measurement is the ScoutPro tear osmolarity system. The ScoutPro is an advanced diagnostic tool designed to quantify tear film osmolarity, a critical biomarker in the pathophysiology of DED. Elevated tear film osmolarity is recognized as a central etiology of DED and is known to correlate with DED severity. The ScoutPro’s automated, rapid assay delivers quantitative osmolarity results in milliosmoles per liter (mOsm/L), aiding clinicians in diagnosing DED and objectively assessing disease progression. The system utilizes a test card with a microfluidic channel to collect a tiny tear sample (50 nanoliters) using passive capillary action. Gold electrodes embedded in the channel then measure the electrical impedance of the tear fluid, which is used to calculate and display the osmolarity result. While tear osmolarity may not differentiate DED subtypes, its ability to classify DED severity with accuracy makes it invaluable for diagnosis, including 71% specificity and 64% sensitivity for the device according to a clinical study of 140 patients from Trukera Medical. By targeting this fundamental biomarker, the device facilitates early diagnosis and personalized treatment strategies to mitigate the inflammatory feedback loop in DED [162,163]. A crucial limitation to this device is that it does not aid in diagnosing DED subtype, as it only measures tear osmolarity. This device needs to be used with other metrics and data to effectively determine the most optimal method of treatment for each individual patient.

**Table 2 ijms-26-08556-t002:** Summary of common diagnostic devices and their utility for DED.

Device	Primary Function	Biomarkers/Parameters Measured	Role in DED Diagnosis
**InflammaDry**	Immunoassay for inflammation detection	MMP-9	Detects elevated MMP-9 levels (>40 ng/mL). High sensitivity and specificity for rapid, in-clinic diagnosis [152].
**I-Pen**	Tear osmolarity system	Tear osmolarity	Measures osmolarity using electrical impedance of the tear fluid of the palpebral conjunctiva [127,128]
**Brill**	Esthesiometry	Corneal sensitivity	Quantifies corneal sensitivity to aid in early detection of corneal dysesthesia and monitoring of treatment efficacy [129].
**ScoutPro**	Tear osmolarity system	Tear osmolarity	Measures osmolarity using microfluidics to collect a tiny tear sample for measurement of electrical impedance of the tear fluid, which is used to calculate the osmolarity result with accuracy [162,163]
**Corneal Topography**	Maps corneal surface to detect irregularities	Corneal surface irregularities, tear film instability	Identifies corneal changes due to tear film instability, enhancing diagnostic precision for DED related ocular surface damage
**Anterior Segment OCT**	High-resolution imaging of anterior chamber structures	Tear film thickness, corneal epithelium, meibomian gland structure	Visualizes alterations in tear film and glands, correlating with DED severity and aiding in diagnosis
**KOWA DR-1a Interferometer**	Analyzes tear film lipid layer dynamics	Lipid layer thickness, tear film stability	Assesses evaporative DED by evaluating lipid layer dynamics, providing insights into tear film instability [164].

While tear osmolarity is lauded as a strong biomarker for DED detection and severity assessment, it is important to acknowledge its limitations. Notably, osmolarity testing alone cannot differentiate between the two primary categories of DED: aqueous-deficient dry eye and evaporative dry eye. While it excels at identifying the presence and overall severity of DED, it is insufficient for determining the specific underlying etiology or for guiding subtype-targeted therapies. Consequently, tear osmolarity measurement should always be utilized alongside other clinical evaluation methods. This highlights the necessity of a comprehensive diagnostic approach, where osmolarity provides a crucial objective measure within a broader assessment framework.

### 4.3. Corneal Esthesiometry

Corneal esthesiometry is used to measure the sensitivity of the corneal nerves by applying a controlled stimulus to the corneal surface. The patient’s involuntary reflex, such as a blink, or subjective response to the stimulus is used to quantify the nerve’s responsiveness. The information gathered from this test helps clinicians in differentiating DED from other ocular surface diseases, assessing DED disease severity based on the observed reduction in corneal sensitivity, and in guiding treatment decisions based on the observed nerve function in individual patients.

The current, most widely used method of measuring corneal sensitivity is with the Cochet-Bonnet Esthesiometer. It is largely considered the gold standard for contact-based esthesiometry and has been long in use by clinicians. The device consists of a handle and a retractable, fine nylon monofilament. The Cochet-Bonnet is used to determine the minimum force of the filament required for a patient to feel a sensation on their cornea. When the patient blinks or reacts to the filament, the length is recorded as the objective measurement for corneal sensitivity. The main benefits of this device are the simplicity, low cost, and portability. It provides a quick and direct measure of nerve function without requiring specialized equipment. However, some limitations exist with this device, as well. Its invasive, contact-based method can cause patient discomfort, induce a reflexive blink, and may potentially lead to a corneal abrasion. The results can also be subjective, as it is easily influenced by the operator’s technique and the patient’s subjective response, making it less objective and reproducible than non-contact devices.

The other device available to clinicians for measuring corneal sensitivity is the Brill Corneal Esthesiometer. This is a novel, non-invasive, handheld device that uses controlled air pulses to measure corneal sensitivity [130]. By measuring a patient’s response to brief corneal stimulation, corneal sensitivity test provides valuable insight into the integrity of corneal nerves and their interaction with the ocular surface [130]. Unlike traditional contact-based esthesiometers, like the Cochet-Bonnet, the Brill esthesiometer employs controlled air pulses to stimulate the cornea, ranging from 2 to 10 mbar across five intensity levels, ensuring precise and reproducible measurements [132]. This non-contact approach minimizes patient discomfort and eliminates the risk of corneal abrasion, making it suitable for use in infectious corneal pathologies. Corneal sensitivity, mediated by the ophthalmic branch of the trigeminal nerve, is often compromised in DED and other conditions like neurotrophic keratopathy, reflecting underlying nerve dysfunction [133]. Corneal sensitivity testing provides valuable insights into the integrity of the corneal nerves, which is essential for diagnosing and managing DED and other ocular surface disorders. The Brill esthesiometer is particularly useful for early detection of DED, monitoring treatment efficacy over time, and providing objective sensitivity data that complements other diagnostic findings. Studies have shown good agreement with the Cochet-Bonnet esthesiometer in healthy and DED patients, though values are not interchangeable, underscoring its role as a complementary diagnostic tool [131]. Additionally, the Brill esthesiometer carries a higher cost and more limited availability in comparison to the Cochet-Bonnet esthesiometer. Additionally, the measurement of corneal sensitivity is not a standalone diagnostic value, as increased sensitivity could be a feature of the different subtypes of DED and even other conditions that impact the ocular surface. Overall, by providing objective data on corneal nerve function, however, the Brill esthesiometer supports tailored therapeutic interventions and improves the management of ocular surface disorders.

### 4.4. Other Imaging Tools

Advancements in biomarker detection technologies are enhancing our ability to study DED’s molecular mechanisms. In ophthalmology, instruments like corneal topography and optical coherence tomography (OCT) are widely used and reliable for assessing DED-related changes. Corneal topography maps the corneal surface to detect irregularities caused by tear film instability, while anterior segment OCT provides high-resolution imaging of the tear film, corneal epithelium, and meibomian glands, revealing structural alterations linked to DED. These imaging modalities enhance diagnosis. The KOWA DR-1α interferometer uses white light illumination to analyze tear film lipid layer dynamics, providing insights into tear film instability. It has a relatively high sensitivity and specificity for diagnosing dry eye, particularly when measuring non-invasive tear break-up time. It is typically used as a complementary device in clinic to help diagnose DED type [164].

### 4.5. Unmet Needs

The gap between biomarker discovery and clinical translation in DED remains a critical challenge, driven by the complexity of translating molecular insights into practical diagnostic tools. While biomarkers like MMP-9, TNF-α, and lactotransferrin have contributed to our understanding of DED’s inflammatory and cellular mechanisms, the integration of some of them into routine clinical practice is hindered by technical, logistical, and economic barriers. MMP-9 has largely broken through these barriers, becoming a strong DED biomarker that is accessible for in-clinic use. However, despite a strong understanding of TNF-α and its role in DED, assessment of it requires more development to become widely adopted in clinical assessment procedures. Similarly, lactrotransferrin has a strong research basis for its relevance in DED but lacks a rapid and reliable method of assessment. Variability in biomarker expression across DED subtypes and patient demographics further complicates the development of standardized assays. This review of current findings demonstrates a need for a device that can reliably test more biomarkers, providing clinicians with a more comprehensive tool for diagnosis of DED.

## 5. Current Treatment Methods and Limitations

The DEWS TFOS III Report outlines current treatments for dry eye based on the etiologies of the disease. The first line of treatment for all types of DED symptoms is typically artificial tears. Artificial tears, however, provide only temporary relief and do not treat the underlying cause of the DED itself. For evaporative DED, eyelid treatment for blepharitis and lid hygiene are effective. These include warm compresses, lid hygiene, and in-office procedures such as thermal pulsation or Intense Pulsed Light (IPL), which can unblock the glands and improve lipid quality in the tears. Recent advancements in lipid-based artificial tears also directly target the tear film’s oily layer, and use of perfluorohexyloctane ophthalmic solution has been shown to help in patients living with evaporative DED. However, the at-home and in-office procedures are limited by efficacy and are not a cure for the disease, meaning repeated treatments are required to maintain relief. The in-office treatments can often be expensive, especially for multiple treatment rounds. Additionally, while the lipid-based tears are more effective at restoring some of the tear film’s lipid layer, it still lacks the ability to resolve the underlying meibomian gland dysfunction that is responsible for the DED in the first place. For aqueous deficient DED, the goal is to increase tear production and conserve existing tears. Treatments include preservative-free artificial tears, prescription anti-inflammatory eye drops (e.g., cyclosporine, lifitegrast) to stimulate tear production, and punctal plugs to block tear drainage and keep tears on the ocular surface for a longer period. While effective in many cases, cyclosporine and lifitegrast are slow acting and can sometimes be uncomfortable for patients to use, leading to poor patient adherence. Punctal plugs are limited by epiphora in certain cases and can cause irritation and foreign body sensation in rare cases. New treatments on the horizon include reproxalap, a reactive aldehyde species that has been shown to reduce inflammation associated with DED by a recent randomized, double-masked, vehicle-controlled dry eye chamber trial of 132 patients from Aldeyra Therapeutics [165,166]. Reproxalap was well tolerated and significantly reduced DED symptoms in patients compared to a vehicle control. Another new, recently FDA-approved treatment is acoltremon, which is a TRPM8 thermoreceptor agonist that has been shown in the COMET studies to be safe and effective in treating DED. These treatments and many others in development may improve efficacy and outcomes in future patients suffering from DED.

## 6. Future Directions

Future advancements in DED diagnosis hinge on the integration of multi-omics approaches, artificial intelligence (AI), and point-of-care devices to enhance precision and accessibility. Emerging technologies, such as exosome profiling and microRNA sequencing, are promising in identifying novel biomarkers that capture the heterogeneity of DED subtypes, potentially enabling personalized treatment strategies. However, significant roadblocks persist, including high costs and limited availability of advanced diagnostic tools in resource-constrained settings, hindering global adoption. For integration of multi-omics into point-of-care diagnostics to be made more feasible, development needs to be done to lower the cost and increase the availability for more widespread use in clinics. Given the extent of sample collection required and high cost, multi-omics remains largely unfeasible at this present moment. Alternatively, AI-driven diagnostic platforms could analyze complex biomarker datasets alongside imaging modalities like corneal topography and OCT, improving diagnostic accuracy and predicting disease progression. It could aid in interpretation of imaging and potentially predict disease progression. However, the challenges of regulatory approval and data privacy remain, and integrating this technology into resource-limited settings could be challenging given high costs and current limited technical support. Standardization of biomarker assays across diverse populations remains challenging due to variability in environmental, genetic, and lifestyle factors. Additionally, the need for large-scale clinical validation studies slows the translation of novel biomarkers into routine clinical practice, necessitating collaborative efforts to establish universal diagnostic criteria and affordable technologies.

Exosomes have emerged as a promising marker for DED, offering potential benefits in both early detection and monitoring of the condition. These nano-sized extracellular vesicles, particularly those found in tear fluid, have shown remarkable potential in identifying DED-specific biomarkers. When analyzed in large scale clinical trials using advanced techniques such as proteomics and RNA sequencing, exosomes from tear samples can reveal distinct molecular signatures associated with various stages and subtypes of DED [167]. Studies have demonstrated significant differences in exosome profiles between healthy individuals and those with DED, including variations in protein content and microRNA expression. Mechanistically, these exosomes can provide valuable insights into the underlying pathological processes of DED, such as inflammation and lacrimal gland dysfunction. Furthermore, exosomes play a crucial role in intercellular communication within the ocular surface ecosystem, making them excellent candidates for monitoring disease progression and treatment response in DED patients.

## 7. Conclusions

DED arises from a complex interplay of inflammation, tear film dysfunction, and cellular responses. Molecular biomarkers such as MMP-9, TNF-α, lactotransferrin, tear osmolarity, exosomes, and miRNAs provide critical insights into the molecular mechanisms underlying the disease. The integration of advanced diagnostic tools, including the KOWA DR-1α interferometer, InflammaDry, ScoutPro, and Brill esthesiometer, with molecular profiling techniques, enables precise diagnosis and monitoring of DED subtypes. These innovations enhance our ability to correlate molecular data with clinical findings, paving the way for personalized therapeutic strategies. By bridging molecular insights with cutting-edge technologies, biomarker-driven research continues to unravel the complexities of DED, fostering global advancements in ocular surface health and driving scientific discoveries toward improved patient outcomes.

## Figures and Tables

**Figure 1 ijms-26-08556-f001:**
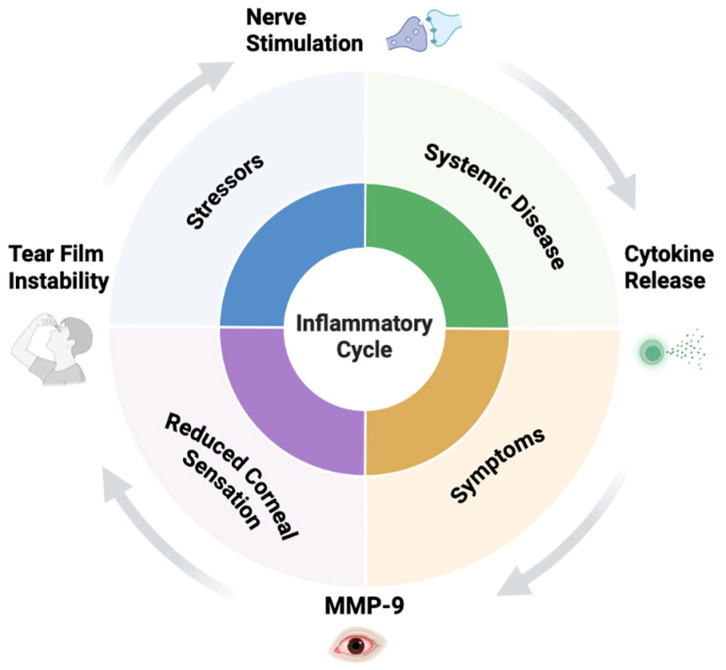
Pathophysiology of DED. This figure illustrates the pathological mechanisms underlying the inflammatory cycle of DED, emphasizing key contributors such as tear film instability, environmental or cellular stress, and systemic conditions. The cycle initiates with tear film instability, which induces ocular surface stress and nerve stimulation, leading to the release of pro-inflammatory cytokines and the activation of an inflammatory response. Elevated levels of MMP-9 further amplify this response by compromising corneal integrity, thereby perpetuating the cycle of inflammation and ocular surface damage. This schematic was developed by the authors’ own concept.

**Figure 2 ijms-26-08556-f002:**
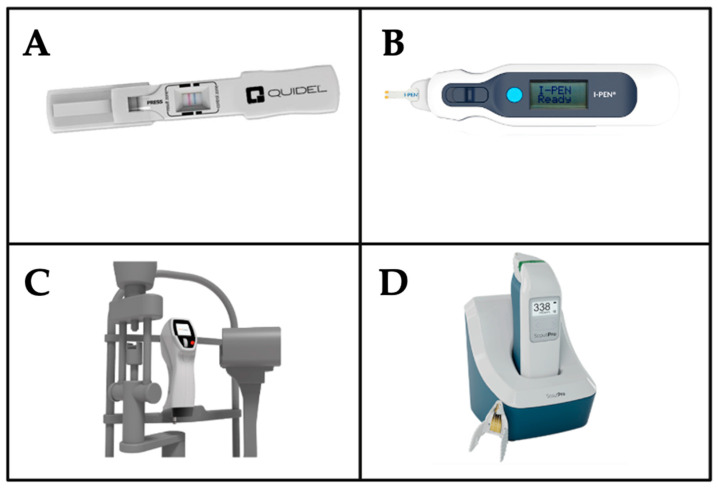
Diagnostic Devices for DED. This figure presents an overview of four distinct medical devices utilized in ophthalmic and diagnostic applications, including (**A**) InflammaDry (MMP-9 detection), (**B**) I-Pen (tear osmolarity), (**C**) Brill Esthesiometer (corneal sensitivity), and (**D**) ScoutPro (tear osmolarity). These tools represent a range of technological innovations aimed at enhancing the accuracy and efficiency of medical assessments in research and clinical settings. Together, these devices provide clinicians with accessible and objective tools for identifying and monitoring biomarkers associated with DED.

## Data Availability

No new data were created in the making of this article.

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
