# Peer review of "Translating Biomarker Discovery: From Bench to Bedside in Dry Eye Disease"

_ijms, 2025, doi:10.3390/ijms26178556_

Round 1
Reviewer 1 Report
Comments and Suggestions for Authors
Interesting study, carefully and well done.
Just one comment: You name certain diagnostic tools/products by its brand name - but you left out many other diagnostic tools. I would leave the brand names away and just discuss the different types of diagnostic tools.
Author Response
Comments 1: Interesting study, carefully and well done. Just one comment: You name certain diagnostic tools/products by its brand name - but you left out many other diagnostic tools. I would leave the brand names away and just discuss the different types of diagnostic tools.
Response 1: Thank you for this suggestion. We agree and think that this improves the overall manuscript. We restructured the other paragraphs to best fit the broad headlines with the descriptions of the tools of interest within the sections. We added in two paragraphs on the Cochet-Bonnet device to make our section on corneal sensitivity more comprehensive. Please see below the specific text that we changed in the manuscript in regards to your comment.
“Corneal esthesiometry is used to measure the sensitivity of the corneal nerves by applying a controlled stimulus to the corneal surface. The patient's involuntary reflex, such as a blink, or subjective response to the stimulus is used to quantify the nerve's responsiveness. The information gathered from this test helps clinicians in differentiating DED from other ocular surface diseases, assessing DED disease severity based on the observed reduction in corneal sensitivity, and in guiding treatment decisions based on the observed nerve function in individual patients.
The current, most widely used method of measuring corneal sensitivity is with the Cochet-Bonnet Esthesiometer. It is largely considered the gold standard for contact-based esthesiometry and has been long in use by clinicians. The device consists of a handle and a retractable, fine nylon monofilament. The Cochet-Bonnet is used to determine the minimum force of the filament required for a patient to feel a sensation on their cornea. When the patient blinks or reacts to the filament, the length is recorded as the objective measurement for corneal sensitivity. The main benefits of this device is the simplicity, low cost, and portability. It provides a quick and direct measure of nerve function without requiring specialized equipment. However, some limitations exist with this device, as well. Its invasive, contact-based method can cause patient discomfort, induce a reflexive blink, and may potentially lead to a corneal abrasion. The results can also be subjective, as it is easily influenced by the operator's technique and the patient's subjective response, making it less objective and reproducible than non-contact devices.” (Lines 529 to 548)
Reviewer 2 Report
Comments and Suggestions for Authors
My congratulation to interesting manuscript that provides a valuable overview of current biomarkers for Dry Eye Disease diagnosis. The inclusion of both molecular and functional biomarkers, such as MMP-9, cytokines, lactotransferrin, and lacritin and tear osmolarity, is particularly useful for clinicians. Diagnostic tools now allow for point-of-care measurement of select biomarkers, including MMP-9 and osmolarity. The integration of advanced diagnostic tools, including the KOWA DR-1α interferometer, InflammaDry, ScoutPro, and Brill esthesiometer, with molecular profiling techniques, enables precise diagnosis and monitoring of DED subtypes. These innovations enhance their ability to correlate molecular data with clinical findings, paving the way for personalized therapeutic strategies.
Comments on the Quality of English LanguageThe English language quality used in the manuscript is clear and appropriate. No major grammatical or language issues were identified.
Author Response
Comment 1: My congratulation to interesting manuscript that provides a valuable overview of current biomarkers for Dry Eye Disease diagnosis. The inclusion of both molecular and functional biomarkers, such as MMP-9, cytokines, lactotransferrin, and lacritin and tear osmolarity, is particularly useful for clinicians. Diagnostic tools now allow for point-of-care measurement of select biomarkers, including MMP-9 and osmolarity. The integration of advanced diagnostic tools, including the KOWA DR-1α interferometer, InflammaDry, ScoutPro, and Brill esthesiometer, with molecular profiling techniques, enables precise diagnosis and monitoring of DED subtypes. These innovations enhance their ability to correlate molecular data with clinical findings, paving the way for personalized therapeutic strategies.
Response 1: Thank you for this comment!
Reviewer 3 Report
Comments and Suggestions for Authors
The manuscript written by Jeremy Jones et al. is a well-structured and informative contribution to the understanding of biomarkers and diagnostic tools in Dry Eye Disease (DED). However, the following issues should be addressed to further strengthen the manuscript:
Comments to the authors
Abstract
- The abstract feels too brief and general. Could the authors expand it by briefly highlighting the diagnostic challenges in DED, specifying key emerging biomarkers and technologies, and clarifying how this review uniquely addresses the translation from bench to bedside?
Introduction
- While the Introduction outlines important aspects of the study, I recommend that the authors refine it by clearly presenting the hypothesis toward the end. Additionally, several statements are made without appropriate references, which makes it difficult to determine whether these are the authors’ own observations or previously reported findings. Including recent and relevant literature (last 3–5 years) will also help position the study within the current scientific context.
Pathophysiology of DED
- While references [2–8] are cited for general DED mechanisms, several specific statements (e.g., impaired lacrimal and meibomian glands, neurosensory feedback) do not have direct citations. Adding recent, high-quality references would improve credibility and help readers verify the described mechanisms.
- For Figure 1, it would be helpful to clarify whether the schematic represents the authors’ own concept or is adapted from other sources. If it is based on previously published work, appropriate references should be included in the figure legend.
- The vicious cycle is a key component in the pathophysiology of DED. The authors are encouraged to explain this cycle in more detail and provide appropriate references to support each step.
Molecular Biomarkers of DED
- In line 151, the term “DED” is abbreviated again, even though it has already been defined earlier in the manuscript. Authors should avoid redundant re-abbreviation and ensure that all abbreviations are used consistently throughout the manuscript.
- In the section discussing miRNAs (lines 277–286), the authors describe the role of miR-204 and other miRNAs in DED inflammatory processes, but no references are provided. Additionally, it is recommended to carefully check the entire manuscript for other instances where references may be missing and ensure all claims are properly supported.
- The manuscript would benefit from a more comprehensive literature search for both clinical and preclinical studies related to each molecular biomarker discussed in each heading. Adding relevant references and data in each section would strengthen the scientific rigor and provide readers with a clearer understanding of the current state of knowledge in DED biomarkers.
- Reference 51, 57, and 33 appear to be outdated. The authors are encouraged to replace it with a more recent reference if available. If the cited work is essential, it can be retained; otherwise, a more up-to-date source would be preferable.
From Biomarkers to Clinical Diagnosis
- In Figure 2, the panel labelling (A, B, C, D) appears inconsistent and unclear. The authors should ensure that each panel is clearly labelled.
- Table 2 does not include references for the data presented. The authors should add appropriate references to ensure scientific accuracy and allow readers to verify the sources.
- For imaging tools (corneal topography, OCT, KOWA DR-1α), could the authors provide more details on sensitivity, reproducibility, and practical application in DED management? Are these tools validated for routine clinical use or mainly research purposes?
- Some devices (InflammaDry, I-Pen, ScoutPro) are described in varying levels of detail. The manuscript could benefit from a consistent format: principle of measurement → clinical utility → limitations →
- The manuscript notes that InflammaDry provides only a positive/negative result, not a quantifiable MMP-9 concentration. Could the authors discuss potential clinical implications of this limitation, e.g., for monitoring treatment response or mild DED cases?
- The manuscript mentions the limitations of the I-Pen, particularly regarding its accuracy and precision. It would be helpful if the authors could briefly discuss why the I-Pen’s performance differs from other tear osmolarity devices, such as TearLab and Wescor, to provide readers with a clearer understanding of their comparative utility in clinical practice.
- Could the authors kindly clarify whether these diagnostic devices are typically used in a complementary manner in clinical practice, or if certain devices are preferred for assessing specific subtypes of DED?
- The manuscript briefly mentions some clinical studies for these devices. Could the authors consider providing additional details on study design, patient populations, and limitations, if available? Including more relevant clinical studies across all diagnostic parameters would strengthen the manuscript and provide a more comprehensive evidence base.
Unmet needs
- Could the authors elaborate on which specific biomarkers are closest to clinical translation, and which remain mostly at the research stage? This would help contextualize the translational gap discussed in Section 4.6.
Future directions
- While AI and multi-omics approaches are highlighted as promising, could the authors comment on the feasibility of implementing these technologies in routine clinical practice, especially in resource-limited settings?
- The discussion on exosomes is informative. Could the authors provide references to recent clinical or preclinical studies demonstrating exosome profiling in DED patients?
Other comments
- The authors are kindly requested to briefly include a section on current treatment strategies for DED, along with their limitations, to provide a more comprehensive context for the manuscript.
Thank you and good luck!
Author Response
Comment 1: The abstract feels too brief and general. Could the authors expand it by briefly highlighting the diagnostic challenges in DED, specifying key emerging biomarkers and technologies, and clarifying how this review uniquely addresses the translation from bench to bedside?
Response 1: To help advance the level of detail in our abstract, we followed the format of this suggestion to specify challenges and emerging technologies in the field, all while trying to maintain brevity. Below is the list of major changes made:
“It can be sometimes elusive to identify the type of DED in patients, given the overlapping symptoms with other conditions like allergies and the multitude of stimuli that might trigger DED onset. There is also difficulty due to limitations on the diagnostic testing available to clinicians, as poor reliability and a lack of standardization plague accurate diagnoses. Identified biomarkers can help identify DED pathophysiology and category, and these include molecular biomarkers like matrix metalloproteinase-9 (MMP-9), cytokines, lactotransferrin, and lacritin, as well as functional biomarkers such as tear osmolarity.” (lines 22 to 28)
“This review uniquely combines a review on the current understanding of various biomarkers with an overview of the emerging technologies available to healthcare providers, aiding in better-informed diagnosis and treatment of DED.” (lines 31 to 34)
Comment 2: While the Introduction outlines important aspects of the study, I recommend that the authors refine it by clearly presenting the hypothesis toward the end. Additionally, several statements are made without appropriate references, which makes it difficult to determine whether these are the authors’ own observations or previously reported findings. Including recent and relevant literature (last 3–5 years) will also help position the study within the current scientific context.
Response 2: Thank you for this critique, I added in the appropriate references in the text and added in a more emphasized hypothesis at the end, displayed below:
“An improved understanding DED-relevant biomarkers and how to best use the tools currently available to quantify these biomarkers will aid in advancing more accurate diagnosis of DED in patients, leading to treatment specific to DED subtype and improved outcomes in the future.” (lines 63 to 67).
Comment 3: While references [2–8] are cited for general DED mechanisms, several specific statements (e.g., impaired lacrimal and meibomian glands, neurosensory feedback) do not have direct citations. Adding recent, high-quality references would improve credibility and help readers verify the described mechanisms.
Response 3: Thank you for this feedback. We agree that citing the specific references for the individual statements made best supports the text written. We cited the specific text referenced to each relevant statement throughout the paragraph, within lines 69 to 93.
Comment 4: For Figure 1, it would be helpful to clarify whether the schematic represents the authors’ own concept or is adapted from other sources. If it is based on previously published work, appropriate references should be included in the figure legend.
Response 4: We agree that this is an important clarification to make. This figure is our own, and line 149 contains a brief statement at the end of the figure caption to let the reader know this, shown below:
“This schematic was developed by the authors’ own concept.”
Comment 5: The vicious cycle is a key component in the pathophysiology of DED. The authors are encouraged to explain this cycle in more detail and provide appropriate references to support each step.
Response 5: We developed this section more to better address this comment. More detail certainly helps to describe the vicious cycle more appropriately to the reader, and we thank the reviewer for bringing this to our attention. See below the specific sentences adjusted to improve on our original paragraph.
“Often, once tear instability presents as the central initiating event, beginning with a breakdown in tear film. This is caused either by a decrease in tear secretion, common in aqueous-deficient DED, or by an increase in tear evaporation, more common in evaporative DED. Once the tear film is compromised, the remaining tear fluid is more concentrated with salts and other solutes, leading to a state of hyperosmolarity.” (Lines 69 to 74)
“This inflammatory cascade is part of a normal bodily inflammatory response, but the chronic presence of these cytokines result in further ocular surface damage and worsens the tear film’s state of dysregulation.” (Lines 80 to 81)
“This further destabilizes the tear film and reinforces the inflammatory environment of the ocular surface.” (Lines 85 and 86)
“Neurosensory abnormalities resulting from this inflammation, such as impairment of corneal nerve function and increased nociceptive signaling, can further lead to significant patient discomfort from increased nociceptive signaling and impaired nerve function [7].” (Lines 86 to 89)
Comment 6: In line 151, the term “DED” is abbreviated again, even though it has already been defined earlier in the manuscript. Authors should avoid redundant re-abbreviation and ensure that all abbreviations are used consistently throughout the manuscript.
Response 6: Thank you for pointing out this oversight. We corrected this to follow the format of the other abbreviations, and this can be found in line 233.
Comment 7: In the section discussing miRNAs (lines 277–286), the authors describe the role of miR-204 and other miRNAs in DED inflammatory processes, but no references are provided. Additionally, it is recommended to carefully check the entire manuscript for other instances where references may be missing and ensure all claims are properly supported.
Response 7: We have added in references to best support this section. (Line 418)
Comment 8: The manuscript would benefit from a more comprehensive literature search for both clinical and preclinical studies related to each molecular biomarker discussed in each heading. Adding relevant references and data in each section would strengthen the scientific rigor and provide readers with a clearer understanding of the current state of knowledge in DED biomarkers.
Response 8: Thank you for this critique, we agree that this would help to bolster the scientific rigor within this manuscript. We added in a few clinical studies throughout this section, highlighted below:
“Prior clinical studies have supported the implication of MMP-9 in DED patients. One study recruited forty-six patients with newly diagnosed DTS and 18 control subjects. They collected 1 microliter of unstimulated tear fluid from each patient and analyzed them for MMP-9 levels. The study found that MMP-9 was significantly higher in DED patients than in the control patients, and the MMP-9 levels even showed to be significantly correlated to DED patients in this study.” (Lines 200 to 205)
“A recent extension study to the Dry Eye Assessment and Management (DREAM) trial sought to find if discontinuation of Omega-3 supplementation in patient previously given Omega-3 as part of the main DREAM study would yield different outcomes in symptoms and discomfort. 22 patients were randomized to Omega-3 supplements and 21 were given a placebo. The results of the study showed that there was no significant difference in symptom outcomes in the group continuing to take the supplement and the group that discontinued it [155]. This study calls into question the significance that Omega-3 fatty acid supplementation has on improving DED. The study is limited by the small cohort size, however, and further research needs to be done to fully evaluate the impact Omega-3 fatty acids may have in DED.” (Lines 374 to 384)
Comment 9: Reference 51, 57, and 33 appear to be outdated. The authors are encouraged to replace it with a more recent reference if available. If the cited work is essential, it can be retained; otherwise, a more up-to-date source would be preferable.
Response 9: This is an excellent point and a great opportunity for us to review the resources we pulled from to ensure that the information is more accurate and up to date. We replaced reference 33 with a newer source, shown in Line 938. However, we determined that 51 and 57 still had crucial information that helped form our writing in the manuscript. Additionally, given that these sources are also backed up by a few other, more recent sources, we feel that retaining sources 51 and 57 is appropriate.
Comment 10: In Figure 2, the panel labelling (A, B, C, D) appears inconsistent and unclear. The authors should ensure that each panel is clearly labelled.
Response 10: This is a great point. We edited and improved the panel labelling to appear more consistent and clear. We think this makes the table appear better as well. See Figure 2 for the changes made.
Comment 11: Table 2 does not include references for the data presented. The authors should add appropriate references to ensure scientific accuracy and allow readers to verify the sources.
Response 11: To fix this and to ensure scientific accuracy, we added in the appropriate references where required. (Table 2)
Comment 12: For imaging tools (corneal topography, OCT, KOWA DR-1α), could the authors provide more details on sensitivity, reproducibility, and practical application in DED management? Are these tools validated for routine clinical use or mainly research purposes?
Response 12: We added in additional details on these devices as suggested. We described how the OCT is widely used in clinics and added in additional details on the sensitivity and specificity for the KOWA DR-1a device, shown below.
“It has a relatively high sensitivity and specificity for diagnosing dry eye, particularly when measuring non-invasive tear break-up time. It is typically used as a complementary device in clinic to help diagnose DED type.” (Lines 591 to 593)
Comment 13: Some devices (InflammaDry, I-Pen, ScoutPro) are described in varying levels of detail. The manuscript could benefit from a consistent format: principle of measurement → clinical utility → limitations →
Response 13: Thank you for this comment. Making our descriptions of each device more consistent and streamlined much improves the information available to the reader in these sections. We added more detail to the ScoutPro and Brill subsections, and added in an additional paragraph describing the Cochet-Bonnet device to provide a more comprehensive overview of tools available. We reorganized these sections to focus on the metrics measured by clinicians (ex. Corneal sensitivity, tear osmolarity, etc.) and described the tools available under these sections. The new text added in is attached below:
“Corneal esthesiometry is used to measure the sensitivity of the corneal nerves by applying a controlled stimulus to the corneal surface. The patient's involuntary reflex, such as a blink, or subjective response to the stimulus is used to quantify the nerve's responsiveness. The information gathered from this test helps clinicians in differentiating DED from other ocular surface diseases, assessing DED disease severity based on the observed reduction in corneal sensitivity, and in guiding treatment decisions based on the observed nerve function in individual patients.The current, most widely used method of measuring corneal sensitivity is with the Cochet-Bonnet Esthesiometer. It is largely considered the gold standard for contact-based esthesiometry and has been long in use by clinicians. The device consists of a handle and a retractable, fine nylon monofilament. The Cochet-Bonnet is used to determine the minimum force of the filament required for a patient to feel a sensation on their cornea. When the patient blinks or reacts to the filament, the length is recorded as the objective measurement for corneal sensitivity. The main benefits of this device are the simplicity, low cost, and portability. It provides a quick and direct measure of nerve function without requiring specialized equipment. However, some limitations exist with this device, as well. Its invasive, contact-based method can cause patient discomfort, induce a reflexive blink, and may potentially lead to a corneal abrasion. The results can also be subjective, as it is easily influenced by the operator's technique and the patient's subjective response, making it less objective and reproducible than non-contact devices.” (lines 529 to 548)
“Corneal sensitivity testing provides valuable insights into the integrity of the corneal nerves, which is essential for diagnosing and managing DED and other ocular surface disorders. The Brill esthesiometer is particularly useful for early detection of DED, monitoring treatment efficacy over time, and providing objective sensitivity data that complements other diagnostic findings.” (lines 568 to 573)
“Additionally, the Brill esthesiometer carries a higher cost and more limited availability in comparison to the Cochet-Bonnet esthesiometer. The measurement of corneal sensitivity is not a standalone diagnostic value, as increased sensitivity could be a feature of the different subtypes of DED and even other conditions that impact the ocular surface.” (Lines 575 to 579)
Comment 14: The manuscript notes that InflammaDry provides only a positive/negative result, not a quantifiable MMP-9 concentration. Could the authors discuss potential clinical implications of this limitation, e.g., for monitoring treatment response or mild DED cases?
Response 14: This is great feedback, we thank you again for the attention to detail. Discussing the limitations of MMP-9 as a metric is important to provide the most information possible to the reader. Below is the text inserted to help clarify this idea:
“This is a clinical limitation, as some patients with milder DED may yield MMP-9 levels below the detectable threshold on the InflammaDry. Additionally, monitoring treatment response may be limited here too, as the MMP-9 level cannot be directly measured and tracked.” (Lines 432 to 435)
Comment 15: The manuscript mentions the limitations of the I-Pen, particularly regarding its accuracy and precision. It would be helpful if the authors could briefly discuss why the I-Pen’s performance differs from other tear osmolarity devices, such as TearLab and Wescor, to provide readers with a clearer understanding of their comparative utility in clinical practice.
Response 15: To better distinguish the I-Pen from other similar devices, we added more detail to highlight how the I-Pen is different in both good and bad ways:
“The I-Pen is very portable and delivers results rapidly, making it a convenient tool for in-clinic use. Comparatively, other similar devices like the Wescor or TearLab are somewhat less portable. Primarily, the differences between these devices are in the collection and analysis methods and in the accuracy of reported results based on prior studies.” (Lines 518 to 521)
Comment 16: Could the authors kindly clarify whether these diagnostic devices are typically used in a complementary manner in clinical practice, or if certain devices are preferred for assessing specific subtypes of DED?
Response 16: This is an important clarification to make, as it helps to show the strengths and weaknesses in the tools available to clinicians today. Below is the text we added in to help improve this:
“Currently, these devices are typically used in the same patients to gather additional date in patients suspected to have DED, yet they do not have the standalone sensitivity to classify the disease etiology.” (Lines 438 to 440)
Comment 17: The manuscript briefly mentions some clinical studies for these devices. Could the authors consider providing additional details on study design, patient populations, and limitations, if available? Including more relevant clinical studies across all diagnostic parameters would strengthen the manuscript and provide a more comprehensive evidence base.
Response 17: Thank you for this comment. We reinforced the clinical studies used in the subsections to improve the evidence base within the manuscript. We also added additional information on other devices on the market to have a more comprehensive snapshot of the tool available for clinicians today.
“The current, most widely used method of measuring corneal sensitivity is with the Cochet-Bonnet Esthesiometer. It is largely considered the gold standard for contact-based esthesiometry and has been long in use by clinicians. The device consists of a handle and a retractable, fine nylon monofilament. The Cochet-Bonnet is used to determine the minimum force of the filament required for a patient to feel a sensation on their cornea. When the patient blinks or reacts to the filament, the length is recorded as the objective measurement for corneal sensitivity. The main benefits of this device are the simplicity, low cost, and portability. It provides a quick and direct measure of nerve function without requiring specialized equipment. However, some limitations exist with this device, as well. Its invasive, contact-based method can cause patient discomfort, induce a reflexive blink, and may potentially lead to a corneal abrasion. The results can also be subjective, as it is easily influenced by the operator's technique and the patient's subjective response, making it less objective and reproducible than non-contact devices.” (Line 582 to 594)
Comment 18: Could the authors elaborate on which specific biomarkers are closest to clinical translation, and which remain mostly at the research stage? This would help contextualize the translational gap discussed in Section 4.6.
Response 18: This is an excellent suggestion, one that certainly improves the message we were trying to convey. Below we have additional text we added into the paragraph.
“MMP-9 has largely broken through these barriers, becoming a strong DED biomarker that accessible for in-clinic use. However, despite a strong understanding of TNF-α and its role in DED, assessment of it requires more development to become widely adopted in clinical assessment procedures. Similarly, lactrotransferrin has a strong research basis for its relevance in DED but lacks a rapid and reliable method of assessment. Variability in biomarker expression across DED subtypes and patient demographics further complicates the development of standardized assays. This review of current findings demonstrates a need for a device that can reliably test more biomarkers, providing clinicians with a more comprehensive tool for diagnosis of DED.” (lines 551 to 559)
Comment 19: While AI and multi-omics approaches are highlighted as promising, could the authors comment on the feasibility of implementing these technologies in routine clinical practice, especially in resource-limited settings?
Response 19: Thank you for this suggestion. We incorporated the response to this into the current text to better emphasize the feasibility and limitations of these technologies to the reader. Here is what we added in:
“However, significant roadblocks persist, including high costs and limited availability of advanced diagnostic tools in resource-constrained settings, hindering global adoption. For integration of multi-omics into point-of-care diagnostics to be made more feasible, development needs to be done to lower the cost and increase the availability for more widespread use in clinics. Given the extent of sample collection required and high cost, multi-omics remains largely unfeasible at this present moment.” (lines 602 to 608)
“It could aid in interpretation of imaging and potentially predict disease progression. However, the challenges of regulatory approval and data privacy remain, and integrating this technology into resource-limited settings could be challenging given high costs and current limited technical support.” (lines 610 to 614)
Comment 20: The discussion on exosomes is informative. Could the authors provide references to recent clinical or preclinical studies demonstrating exosome profiling in DED patients?
Response 20: We assessed clinical trials that highlight the potential future of exosome profiling to best support the text. The added context is highlighted below:
“When analyzed in large scale clinical trials using advanced techniques such as proteomics and RNA sequencing, exosomes from tear samples can reveal distinct molecular signatures associated with various stages and subtypes of DED.” (lines 622 to 625)
Comment 21: The authors are kindly requested to briefly include a section on current treatment strategies for DED, along with their limitations, to provide a more comprehensive context for the manuscript.
Response 21: Thank you for this suggestion. We feel that this section will greatly improve the overall context of this paper. Below is the text for the section we added in:
“The DEWS TFOS III Report outlines current treatments for dry eye based on the etiologies of the disease. The first line of treatment for all types of DED symptoms is typically artificial tears. Artificial tears, however, provide only temporary relief and do not treat the underlying cause of the DED itself. For evaporative DED, eyelid treatment for blepharitis and lid hygiene are effective. These include warm compresses, lid hygiene, and in-office procedures such as thermal pulsation or Intense Pulsed Light (IPL), which can unblock the glands and improve lipid quality in the tears. Recent advancements in lipid-based artificial tears also directly target the tear film's oily layer, and use of perfluorohexyloctane ophthalmic solution has been shown to help in patients living with evaporative DED. However, the at-home and in-office procedures are limited by efficacy and are not a cure for the disease, meaning repeated treatments are required to maintain relief. The in-office treatments can often be expensive, especially for multiple treatment rounds. Additionally, while the lipid-based tears are more effective at restoring some of the tear film’s lipid layer, it still lacks the ability to resolve the underlying meibomian gland dysfunction that is responsible for the DED in the first place. For aqueous deficient DED, the goal is to increase tear production and conserve existing tears. Treatments include preservative-free artificial tears, prescription anti-inflammatory eye drops (e.g., cyclosporine, lifitegrast) to stimulate tear production, and punctal plugs to block tear drainage and keep tears on the ocular surface for a longer period. While effective in many cases, cyclosporine and lifitegrast are slow acting and can sometimes be uncomfortable for patients to use, leading to poor patient adherence. Punctal plugs are limited by epiphora in certain cases and can cause irritation and foreign body sensation in rare cases. New treatments on the horizon include reproxalap, a reactive aldehyde species that has been shown to reduce inflammation associated with DED by a recent randomized, double-masked, vehicle-controlled dry eye chamber trial of 132 patients from Aldeyra Therapeutics. Reproxalap was well tolerated and significantly reduced DED symptoms in patients compared to a vehicle control. Another new, recently FDA-approved treatment is acoltremon, which is a TRPM8 thermoreceptor agonist that has been shown in the COMET studies to be safe and effective in treating DED. These treatments and many others in development may improve efficacy and outcomes in future patients suffering from DED. “(Lines 561 to 595.)